# The DOF Transcription Factors in Seed and Seedling Development

**DOI:** 10.3390/plants9020218

**Published:** 2020-02-08

**Authors:** Veronica Ruta, Chiara Longo, Andrea Lepri, Veronica De Angelis, Sara Occhigrossi, Paolo Costantino, Paola Vittorioso

**Affiliations:** Department of Biology and Biotechnology, Sapienza University of Rome, P.le Aldo Moro 5, 00185, Rome, Italy; veronica.ruta@uniroma1.it (V.R.); longo.493622@studenti.uniroma1.it (C.L.); lepri.1689102@studenti.uniroma1.it (A.L.); deangelis.1695787@studenti.uniroma1.it (V.D.A.); occhigrossi.1634553@studenti.uniroma1.it (S.O.); paolo.costantino@uniroma1.it (P.C.)

**Keywords:** DOF proteins, seed germination, DELLA proteins, seed maturation, seedling development

## Abstract

The DOF (DNA binding with one finger) family of plant-specific transcription factors (TF) was first identified in maize in 1995. Since then, DOF proteins have been shown to be present in the whole plant kingdom, including the unicellular alga *Chlamydomonas reinhardtii*. The DOF TF family is characterised by a highly conserved DNA binding domain (DOF domain), consisting of a CX_2_C-X_21_-CX_2_C motif, which is able to form a zinc finger structure. Early in the study of DOF proteins, their relevance for seed biology became clear. Indeed, the PROLAMIN BINDING FACTOR (PBF), one of the first DOF proteins characterised, controls the endosperm-specific expression of the zein genes in maize. Subsequently, several DOF proteins from both monocots and dicots have been shown to be primarily involved in seed development, dormancy and germination, as well as in seedling development and other light-mediated processes. In the last two decades, the molecular network underlying these processes have been outlined, and the main molecular players and their interactions have been identified. In this review, we will focus on the DOF TFs involved in these molecular networks, and on their interaction with other proteins.

## 1. Introduction

The first DOF (DNA binding with one finger) proteins were isolated because of their interaction with viral or bacterial sequences [1,2,3]. Indeed, the maize DOF1/MNB1a was originally identified as a protein binding to the cauliflower mosaic virus 35S (CaMV35S) promoter [1,4]; similarly, the *Arabidopsis* OBP1 (ocs Binding Factor/OBF BINDING PROTEIN 1) was shown to bind an element upstream of the *ocs* domain present in the CaMV35S promoter [2]. The tobacco NtBBF1 (*Nicotiana tabacum rolB* domainB Factor1) DOF protein was identified as the plant transcription factor binding to the promoter of the plant oncogene *rolB* [3], where it recognises a specific sequence in the region required for *rolB* expression in root meristematic cells and for induction by auxin [5,6]. All DOF proteins bind the highly conserved (T/A)AAAG consensus motif, identified through binding site-selection experiments using the maize DOF proteins DOF1, DOF2, DOF3 and PBF (PROLAMIN BINDING FACTOR) [7]. Although these first DOF proteins have been linked to viral and bacterial activity, subsequent studies highlighted their fundamental role in plant-specific processes (for a review, see References [8,9,10]).

In this review, we will focus on the role of DOF proteins in seed and seedling developmental processes.

In *Arabidopsis*, seed development is divided in two phases: embryo/endosperm development and seed maturation [11]. Once the embryo is formed and cell division arrests, the seed enters the maturation phase characterised by an increase of the levels of abscisic acid (ABA) required for the establishment and maintenance of dormancy upon completion of maturation. Germination occurs when seeds are in optimal environmental conditions, mainly as far as water availability, light and temperature [12,13], and dormancy is released—under experimental conditions, dormancy can be released by a period of storage and a cold treatment at 4 °C for two days (stratification). The positive effect of light on seed germination is mediated mainly by the Red-light photoreceptor phytochrome B (phyB) [14], which controls the balance between ABA, which promotes dormancy, and gibberellic acid (GA), which stimulates germination by counteracting the effect of ABA [15,16,17]. Once germination is completed, seedling development undergoes photo- or skoto-morphogenesis, depending on the presence or absence of light, respectively [18]. An in-depth analysis of these processes is covered in other articles of this issue.

## 2. DOF TFs Regulate Seed Storage Protein Accumulation and Mobilisation

In cereal seeds, Prolamins are the main class of storage proteins in starchy endosperm cells. They are characterised by a region rich in proline and glutamine and are synthesised in the endoplasmic reticulum during seed development. The expression of prolamin genes is tightly controlled. The main cis-element present in their promoters is an endosperm-specific box [19,20], which consists of two motifs: a GLM (GCN4-like motif) (5′ G(A)TGA(G) GTCAT 3′) that shares homology with yeast GCN4 [21], and a 7 bp P-box (Prolamin box) (5′TGTAAAG3′) [22,23,24]. The endosperm nuclear factors binding the P-box on the promoters of barley prolamin genes were among the first DNA binding factors identified in plants [25,26,27,28]. The corresponding *PBF* (*PROLAMIN BINDING FACTOR*) gene was first isolated from maize and shown to encode a DOF protein that interacts with O2 (Opaque2) [29]. The barley and wheat homologues of maize PBF—BPBF and WPBF, respectively—were shown to interact in vitro with the P-box motif present in the promoter of the barley *Hor2* (*Hordein-2*) gene, encoding a barley prolamin [30] (Table 1). The GLM motif is recognised and bound by bZIP (basic leucine zipper) transcription factors of the O2 sub-family [31,32,33,34]; indeed, mutations of the maize *O2* gene result in a endosperm-specific decreased expression of the genes encoding the maize prolamin proteins, named zeins [35,36]. The barley bZIP proteins, BLZ1 and BLZ2, were shown to bind the promoter of the *Hor-2* gene [32,34]. Transient expression experiments in barley endosperm showed that BLZ1 and BLZ2 transactivate a synthetic promoter containing both the GLM and the P-box motifs through homo or heterodimer formation [32,34], and that the presence of the P-box motif was required for this transactivation [34]. Conservation of this DOF-bZIP molecular module is also suggested by the cooperation of the rice PBF orthologs (RPBF) with the bZip transcription factor, RISBZ1, in inducing the expression of the storage protein genes in rice [37].

BPBF has also been shown to interact with GAMYB, a barley transcription factor belonging to the R2R3 MYB family [38], which binds a 5′ AACAAC 3′ element, close to the endosperm-specific box. In barley endosperm, GAMYB cooperates with BPBF to induce expression of the *Hor2* gene [38], and the in vivo interaction between the two proteins has been shown in onion cells by means of the BiFC (bimolecular fluorescent complex) approach [39]. Interestingly, the same authors proved that, besides BPBF, the barley DOF protein, SAD (SCUTELLUM and ALEURONE-expressed DOF), was also able to bind the *Hor2* promoter and to interact in vivo with GAMYB [39]. Accordingly, the simultaneous presence of both these DOF proteins in co-transfection experiments, resulted in an additive trans-activation effect [39].

Besides storage proteins, endosperm also contains starch, which is crucial for both seed yield and quality. The maize ZmDOF36 protein has been recently demonstrated to positively control starch accumulation [40]. Indeed, *ZmDOF36* overexpressing lines showed upregulation of starch biosynthetic genes and increased starch content. In addition, six of these biosynthetic genes, namely *ZmAGPS1a*, *ZmAGPL1*, *ZmGBSSI*, *ZmSSIIa*, *ZmISA1* and *ZmISA3*, were directly bound by ZmDOF36, in a yeast one-hybrid assay [40]. Similarly, it was previously shown that knock-down of the endosperm-specific *ZmDOF3* gene results in reduced starch content in endosperm of the *ZmDOF3* RNAi transgenic lines [41].

Lipids are also important seed storage compounds, mainly in oil crops such as soybean, maize and cotton. The soybean DOF proteins GmDOF4 and GmDOF11 have been shown to directly induce the acetyl CoA carboxylase and long-chain-acyl CoA synthetase biosynthetic genes [42]. Consistently, lipid and total fatty acids content was increased in *GmDOF4* and *GmDOF11* transgenic Arabidopsis seeds [42]. Similarly, overexpression of *GhDOF1* from *Gossypium hirsutum* resulted in an increase of lipid levels in cotton seeds [43].

At the onset of seed germination, stored compounds, mainly proteins and starch, are hydrolysed by proteases and hydrolytic enzymes secreted from aleurone cells [44,45]. GAs induce the expression of these proteases- and hydrolases-encoding genes: the promoters of these genes are characterised by a conserved GA response complex (GARC), comprising three cis-acting motifs: a GA-responsive element (GARE), a DOF binding site and a TATCCAC box [46,47]. GAs also induce transcription of the GAMYB transcription factor, which binds the GARE box [48,49]. GAMYB trans-activation is counteracted by BPBF which, besides inducing prolamin genes, has been shown to repress the *Al21* gene, encoding a cathepsin B-like protease, through binding to the DOF binding site of the GARC [50]. Interestingly, SAD is also involved in the control of *Al21* expression, but with an antagonistic function with respect to BPBF [51]. Indeed, in regulating the expression of hydrolase genes in aleurone cells, BPBF functions as a repressor, whereas SAD functions as a transcriptional activator, although both interact with GAMYB [50,51]. Moreover, two additional DOF proteins, HvDOF17 and HvDOF19, have been shown to negatively control the GAMYB-mediated expression of the *Al21* gene in germinating barley aleurones and to interact with GAMYB. HvDOF19 binds the three DOF binding sites present in the *Al21* promoter to repress *Al21* in an ABA-dependent manner, whereas HvDOF17 functions by reducing the GAMYB affinity for the GARE box, following HvDOF17-GAMYB direct interaction [52] (Table 1).

## 3. Interaction DOF-DELLA Represses Seed Germination

DELLA proteins are repressors of GA signalling and of GA-mediated processes, such as seed germination. In *Arabidopsis*, there are five DELLA-encoding genes, namely *RGA*, *GAI*, *RGL1-3* (*REPRESSOR OF ga1-3*, *GA INSENSITIVE*, *RGA LIKE 1-3*). DELLA repression is relieved following GA binding to the GID1 (GA-INSENSITIVE DWARF1) soluble receptor [53,54]. DELLA proteins are bound by the GA-GID1 complex and are subsequently recognised by the F-box SLY (SLEEPY) protein, which targets DELLAs to degradation via the ubiquitin-proteasome 26S pathway [54,55]. DELLA proteins do not directly bind DNA but act through interaction with the DNA-binding domain of transcription factors, thus blocking their transcriptional activity [56,57,58]. DELLAs can also activate transcription by sequestering inhibitors [59,60].

As for the seed germination process, it was shown that the DELLA protein primarily involved in this process is RGL2 [61,62,63]. A genome-wide comparative study allowed us to identify genes specifically regulated by RGL2. This analysis compared the transcriptome of *ga1-3 rga-t2* seeds—which are not able to germinate—with that of *ga1-3rga-t2rgl2-1*, in which a lack of RGL2 results in the rescue of seed germination [63]. Analysis of the promoters of the upregulated Differentially Expressed Genes (DEGs) revealed a significant enrichment in DOF binding sites, thus suggesting that RGL2 might interact with DOF transcription factors to induce expression of the target genes [63]. Further studies on the RGL2 target gene *GATA12*, encoding a GATA zinc finger transcription factor, led to the identification of the RGL2-DOF6 (At3g45610; DOF3.2/DOF6) complex responsible for *GATA12* induction in freshly harvested seeds. The direct interaction between RGL2 and DOF6, demonstrated by a yeast two-hybrid assay, was confirmed by Co-IP (Co-Immunoprecipitation) in *Nicotiana benthamiana* leaves [64]. GATA12 is a downstream repressor of GA-mediated seed germination, and it contributes to the establishment of seed dormancy. ChIP (Chromatin Immunoprecipitation) assays proved that both RGL2 and DOF6 are required to induce *GATA12* expression [64] (Figure 1).

DOF6 was previously shown to negatively affect seed germination, increasing both the ABA level and the expression of ABA-related genes in seeds [65]. By yeast two-hybrid and bimolecular fluorescent complementation assay in onion cells, it was also demonstrated that DOF6 interacts with the positive regulator of germination TCP14 (TEOSINTE BRANCHED1/CYCLOIDEA/PROLIFERATING CELL FACTOR14), thus suggesting a competitive action of these two transcription factors during dormancy and germination of seeds [65]. TCP14, which has been shown to promote GA-mediated cell proliferation during seed germination, directly interacts with both GAI and RGL2, which inhibit the DNA-binding activity of TCP14, thus inactivating its positive effect on seed germination [66].

Although it was proposed that neither RGA nor GAI plays a major role in controlling GA-dependent seed germination [67], extensive genetic analysis with different combinations of *della* mutant alleles clearly revealed that, in addition to RGL2, RGA and GAI also repress seed germination, and that their function is light-dependent [68]. The molecular mechanism of this control was unveiled by Oh and collaborators [69], who showed that *GAI* and *RGA* are transcriptionally induced in the dark by PIF1 (PHYTOCHROME INTERACTING FACTOR1), the master repressor of seed germination [70]. In the light, PIF1 is degraded via the proteasome 26S and, in turn, the expression of *GAI* and *RGA* is downregulated [69].

Another DOF protein, which represses seed germination through direct interaction with a DELLA protein, is DAG1 (DOF AFFECTING GERMINATION 1), which was convincingly demonstrated to be involved in seed germination [71]. Indeed, *dag1* knockout mutant seeds show reduced dormancy and require lower Red-light fluence rates than wild-type seeds to germinate [71,72]. In addition, *dag1* mutant seeds require about a ten times lower GA concentration than wild-type to reach 50% germination [72]. DAG1 acts in the phyB-mediated pathway, downstream of PIF1, and it controls the ABA/GA ratio by repressing the GA biosynthetic gene *GA3ox1* and the ABA catabolic gene *CYP707A2* [73,74].

DAG1 cooperates with GAI to negatively regulate *GA3ox1*. Indeed, GAI is necessary for the binding of DAG1 to the DOF sites in the *GA3ox1* promoter, and it directly interacts with DAG1 [75]. The cooperation between DAG1 and GAI is further strengthened by the reciprocal control of these two factors, which mutually effect their expression [75] (Figure 1).

GAI has long been believed to have overlapping functions with the other DELLA protein RGA in negatively controlling seed germination and other plant processes [67,69,76]. However, more recently, it was shown that RGA and GAI have distinct roles in this process, since *rga28* mutant seeds have been demonstrated to be more sensitive to stratification, a phenotype probably due to an increased dormancy, thus suggesting that RGA, but not GAI, promotes seed dormancy [77]. In addition, a transcriptomic analysis showed that GAI, but not RGA, is upregulated by *DAG1* inactivation [78], consistently with the reciprocal transcriptional control of *DAG1* and *GAI* during seed germination [75].

Interestingly DAG2, the DOF protein which plays an antagonistic role to DAG1 [72], has been shown to negatively control expression of *RGA*, but not of *GAI*. Indeed, *RGA* transcript level is significantly increased in *dag2* mutant seeds exposed to Red light, consistently with the function of DAG2 as a positive regulator of seed germination in the PIF1-DAG1 molecular pathway [79].

## 4. DOF Proteins in Seedling Development and Other Light-Mediated Processes

Seedling development depends on environmental conditions. Indeed, once germination is completed, seedlings undergo two possible developmental programs, photomorphogenesis or skotomorphogenesis, depending on the presence or absence of light, respectively [18]. Photomorphogenesis is characterised by inhibition of hypocotyl elongation, open and expanded cotyledons and chloroplast development, whereas skotomorphogenesis is characterised by long hypocotyls and small unfolded cotyledons. Light, mainly through the photoreceptors phytochrome A (phyA) and phyB, mediates these developmental programs via downstream signalling molecules, and through the control of hormonal levels. Besides phyA and phyB, the blue/UV-A absorbing cryptochromes (cry) have partially redundant functions in the control of photomorphogenic responses, such as hypocotyl elongation and cotyledon expansion (for a review, see Reference [80]). The PIF bHLH transcription factors are the main downstream signalling factors of the phytochromes. Seedlings of the *pifq* quadruple mutant lacking PIF1, PIF3, PIF4 and PIF5 display a constitutive photomorphogenic phenotype when grown in the dark, thus suggesting that these PIF factors redundantly promote skotomorphogenesis [81,82,83]. This action is counteracted by DELLA proteins, which inhibit PIF activity or promote degradation of PIF factors [84], thus promoting photomorphogenesis. Gas, in turn, promote degradation of DELLA proteins, therefore leading to stabilisation of PIF proteins [57,58]. Indeed, GAs promote etiolated growth and repress photomorphogenesis in the dark [85].

GA metabolism is controlled by light during seedling development, as light induces the expression of the GA catabolic genes in seedlings, thus lowering GA level [86]. GAs and light, which antagonistically control photomorphogenesis, both converge on PIF proteins.

COG1 (COGWHEEL 1) is the first DOF protein which was shown to be involved in both phyA- and phyB-mediated seedling development. The *cog1-D* dominant mutant overexpressing *COG1*, shows attenuated light-mediated responses, namely light-mediated inhibition of hypocotyl elongation, cotyledon opening, anthocyanin accumulation and light-inducible gene expression [87]. Although COG1 is likely a negative regulator in both phyA and phyB signalling pathways, *COG1* expression is induced by both Red and Far Red light, thus suggesting that this DOF factor may be a component of a fine-tuning mechanism in phytochrome signalling [87]. More recently, it has been shown that COG1 promotes hypocotyl elongation, through PIF4 and PIF5, which in turn induce Brassinosteroids (BRs) biosynthesis. Indeed, COG1 directly binds the promoter of *PIF4* and *PIF5* as revealed by the ChIP assay [88].

COG1 is also involved in seed tolerance to deterioration, the process of loss of vigour and viability: indeed, the dominant allele *cog1-2D* that overexpresses *COG1* displayed increased tolerance to deterioration, a phenotype dependent on a higher GA level and a reduced permeability of the seed coat [89]. This phenotype is in agreement with the negative role of COG1 in light signalling, since Red and Far Red light have a negative role in seed tolerance to deterioration, as indicated by the increased tolerance of the *phyB-9* and *phyA-211* mutant seeds [89]. In addition, it has recently been demonstrated that COG1 controls the expression of the peroxidases PRX2 and PRX25, which are involved in the polymerisation of suberin in the seed coat [90]. Also, the Arabidopsis DOF protein, AtDOF4.2 (At4g21030), has been proposed to be involved in seed coat composition and mucilage production [91].

The dominant mutant allele *cdf4-1D* also shows increased seed tolerance to deterioration due to overexpression of *CDF4* (*CYCLING DOF FACTOR 4*) [89]. CDF4 belongs to a sub-family of DOF proteins homologous to CDF1, which negatively controls expression of the floral activator-encoding gene *CO* (*CONSTANS*) [92]. The CDF1-4 proteins redundantly repress the transcription of *CO*, thus representing key elements of the photoperiodic control of flowering in Arabidopsis. Interestingly, CDF and COG1 belong to the same DOF phylogenetic clade, previously referred to as group II [93].

In addition to being involved in the negative control of clock-dependent photoperiodic flowering response [94,95], it was recently shown that CDF5 promotes cell expansion and hypocotyl elongation in a light- and clock-dependent manner [96]. CDF5 acts downstream of both the PIFs and the PRR9/7/5 (PSEUDO-RESPONSE REGULATORS 9/7/5) antagonistic pathway, which promote and, respectively, inhibit hypocotyl growth. Both PIFs and PRRs target CDF5: PIFs induce pre-dawn expression of *CDF5* to promote hypocotyl growth, whereas PRRs directly repress *CDF5* from morning to dusk to prevent overgrowth of the hypocotyl [96] (Figure 2).

Similarly, DAG1 has been recently shown to promote cell expansion and hypocotyl elongation [97]: light-grown *dag1* mutant seedlings have hypocotyls significantly shorter than the wild-type, suggesting that DAG1 is a negative regulator in the light-mediated inhibition of hypocotyl elongation [73]. Through a genome-wide analysis, it was shown that DAG1 promotes hypocotyl elongation through the control of ABA, ethylene and auxin signalling. Indeed, Gene Ontology analysis of the 257 DE (Differentially Expressed) genes in *dag1* hypocotyls compared to the wild-type, revealed that “response to abscisic acid”, “ethylene biosynthetic process” as well as “response to ethylene” and “response to auxin” were among the most significantly enriched categories [97]. Consistently, DAG1 directly binds to the promoters of *WRKY18* that encodes a transcription factor involved in ABA signalling, of the ethylene-induced gene, *ERF2* (*ETHYLENE RESPONSE FACTOR 2*), and of the *SAUR67* (*SMALL AUXIN UP RNA 67*), an auxin-responding gene encoding a protein promoting hypocotyl cell expansion [97].

The Arabidopsis DOF protein, OBP3 (OBF4 BINDING PROTEIN 3), has also been shown to be involved in light-dependent inhibition of hypocotyl elongation. Indeed, overexpression of *OBP3* in the activation-tagged line *sob1-D* (*suppressor of phyB-4 dominant*) reverted the long hypocotyl phenotype of the *phyB*-4 mutant [98], suggesting that OBP3 promotes phyB-mediated inhibition of hypocotyl elongation. Interestingly, reduced *OBP3* expression in the *OBP3-RNAi* lines displayed larger cotyledons, mainly under blue light, indicating that OBP3 acts downstream of both the phyB and cry1 photoreceptors, to repress cell expansion [98].

## 5. DOF in Early Steps of Arabidopsis Development

Arabidopsis *DOF* genes are mainly expressed in the vascular system, in the xylem, in the phloem or both (for a review, see Reference [99]). Comparison of transcriptional and translational profiles suggested that DAG1 and ITD1 (INTERCELLULAR TRAFFICKING DOF1) were possibly capable of moving from the tissue where they were expressed (phloem) to the neighbouring tissue (pericycle), and function as non-cell autonomous (NCA) transcription factors [100]. Experimental evidence of the plasmodesmata-dependent movement from the root stele to the endodermis has been subsequently provided for ITD1 by Chen and co-authors [101]. These authors also identified the intercellular trafficking motif (ITM) that is responsible for selective cell-to-cell movement and encompasses the DOF domain and one element of the bipartite NLS (Nuclear Localisation Signal) [101].

More recently, two Arabidopsis DOF proteins (DOF2.4 and DOF5.1) have been identified as mobile factors involved in root procambial development and were named PEAR1 and 2 (PHLOEM EARLY DOF1 and 2). The corresponding *PEAR1* and *2* genes are highly expressed in protophloem sieve elements (PSE) [102]. PEAR1 and 2 were shown to trigger periclinal cell division by controlling genes that promote radial growth, such as *SUPPRESSOR OF MAX2 1-LIKE3* (*SMXL3*), encoding a key regulator of phloem formation [103]. In addition, four homologues (DOF3.2, DOF5.3, DOF1.1 and DOF5.6 (Table 2), have been identified as PSE-specifically or PSE-abundantly expressed *DOF* genes with a broader protein localisation. These DOF factors redundantly function in the promotion of periclinal cell divisions in PSE cells [102]. The action of PEAR factors is counteracted by the HD-ZIP III transcription factors PHB, CNA and REV (PHABULOSA, CORONA and REVOLUTA). In addition, this molecular network involves a double-negative feedback loop where PEAR1 induces transcription of the *PHB*, *CNA*, and *REV* genes and the corresponding PHB, CNA and REV proteins negatively control PEAR1 transcription and protein movement [102].

The transition from seeds’ dormancy to germination involves a wide reprogramming of the transcriptome and involves chromatin remodelling. The PRC2 (Polycomb Repressive Complex 2), which is responsible for the repressive mark H3K27me3, has been shown to target several master regulators of this transition [108,109]. Indeed, PRC2 is required to switch off these seed-specific genes to allow seedling growth [108]. Consistently, *fie* (*fertilization independent endosperm*) mutant seeds, lacking a functional PRC2, displayed dormancy and germination defects [108]. Genome-wide analysis of the chromatin state of wild-type and *fie* seedlings revealed a number of *DOF* genes (Table 3) as putative targets of PRC2 [108]. Further studies on one of these, *DAG1*, showed that in developing seedlings, its transcribed region is significantly enriched in the H3K27me3 repressive mark [74]. Interestingly, among the *DOF* genes marked by H3K27me3, *PEAR* genes were also found, consistent with their master role during vascular development and root radial growth.

## 6. Conclusions

DOF proteins are plant-specific transcription factors present in the whole plant kingdom, including the green unicellular alga *Chlamydomonas reinhardtii* (1 *DOF* gene), the moss *Physcomitrella patens* (19 *DOF* genes) and the model plant *Arabidopsis thaliana* (36 *DOF* genes) [117,118]. Phylogenetic analysis suggested that *DOF* genes originated from a common ancestor (conserved as a single copy in *Chlamydomonas*) that underwent structural and functional diversification through several rounds of gene duplications [117]. This evolutionary process may be linked with the development of new specific functions required to cope with increasingly complex plant developmental regulatory networks. As a result, DOF factors are involved in diverse plant developmental processes, including seed dormancy, germination and photomorphogenesis. The recent evidence that a number of Arabidopsis DOF factors play a key role in the promotion of radial growth in the procambial tissues of the root apical meristem [102], further emphasises the importance of this gene family in plant life.

## Figures and Tables

**Figure 1 plants-09-00218-f001:**
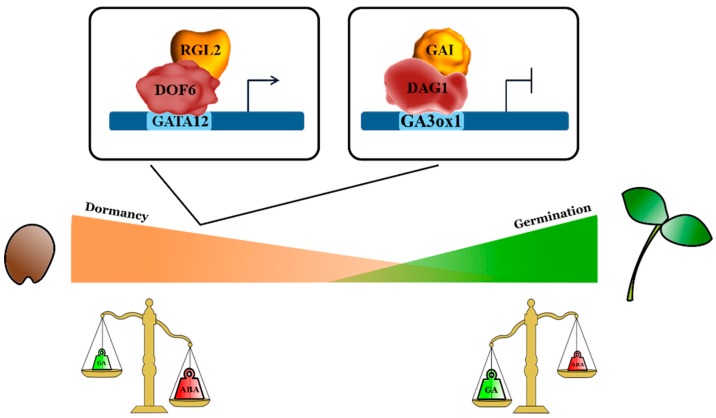
DOF proteins interact with DELLA factors to negatively regulate seed germination. On the left: the *Arabidopsis* DOF6 protein (AT3G45610; DOF3.2) interacts with the DELLA protein RGL2. The complex RGL2-DOF6 induces *GATA12* expression, in freshly harvested seeds, to establish seed dormancy [64]. On the right: the *Arabidopsis* DAG1 protein requires the DELLA protein GAI to repress the GA biosynthetic gene *GA3ox1* during maturation, dormancy and germination of seeds [75]. RGL2 (RGA LIKE2), GAI (GA INSENSITIVE).

**Figure 2 plants-09-00218-f002:**
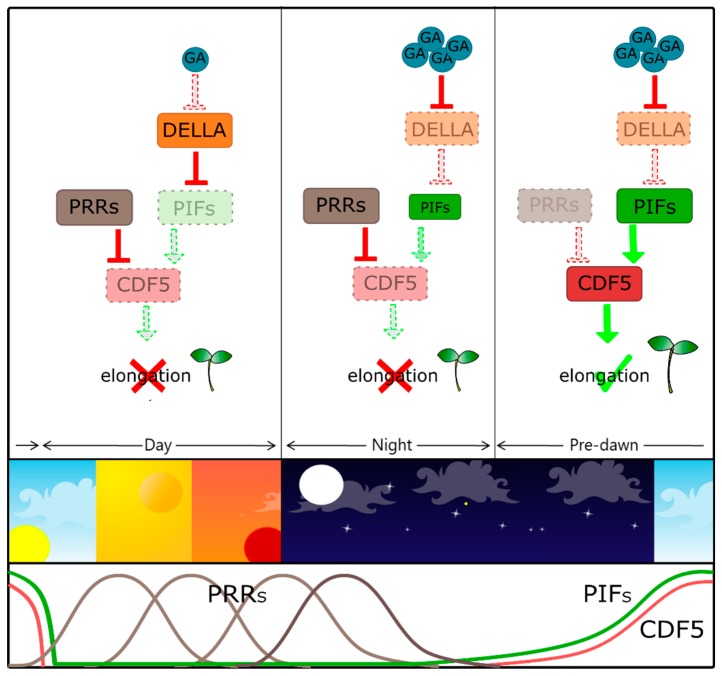
CDF5 promotes hypocotyl elongation in a clock-dependent manner. Expression of *CDF5* is antagonistically regulated by both PIFs and PRRs, which directly interact, in order to control hypocotyl elongation during the light/dark cycles. From morning to dusk: accumulation of DELLA proteins, due to the absence of GAs, represses PIFs. PRR proteins (PRRs) directly repress CDF5, leading to inhibition of hypocotyl elongation. During the night: GAs promote DELLA degradation and in turn, stabilisation of PIFs. PRRs inhibit the activity of PIFs in post-dusk phase, gating the hypocotyl elongation to pre-dawn (upper panel). Schematic representation of the PRRs, PIFs and CDF5 proteins temporal activity (lower panel) [96]. CDF5 (CYCLING DOF FACTOR 5), PRRs (PSEUDO-RESPONSE REGULATORS), PIFs (PHYTOCHROME INTERACTING FACTORS).

**Table 1 plants-09-00218-t001:** *DOF* (DNA binding with one finger) genes involved in seed storage protein accumulation and mobilization.

Name	Species	Function	References
PBF	*Zea mays*	↑ prolamin genes	[29]
BPBF	*Hordeum vulgare*	↑ hordein genes ↓ hydrolase genes	[30,39,50]
SAD	*Hordeum vulgare*	↑ hordein genes ↑ hydrolase genes	[39,51]
HvDof17	*Hordeum vulgare*	↓ hydrolase genes	[52]
HvDof19	*Hordeum vulgare*	↓ hydrolase genes	[52]
WPBF	*Triticum aestivum*	↑ prolamin genes	[30]
RPBF	*Oryza sativa*	↑ prolamin genes	[37]

Reported in the table are the DOF proteins characterised as Prolamin Binding Factors, the species, the function and the references. The up arrow means transcriptional activation, whereas the down arrow means transcriptional repression. PBF (PROLAMIN BINDING FACTOR), BPBF (Barley PROLAMIN BINDING FACTOR), WPBF (Wheat PROLAMIN BINDING FACTOR), RPBF (Rice PROLAMIN BINDING FACTOR), SAD (SCUTELLUM and ALEURONE-expressed DOF).

**Table 2 plants-09-00218-t002:** Arabidopsis *DOF* genes encoding PHLOEM EARLY DOF (PEAR) proteins.

Name	Gene ID	Synonymous	References
PEAR1	AT2G37590	AtDof2.4	[102]
PEAR2	AT5G02460	AtDof5.1	[102,104]
DOF6	AT3G45610	AtDof3.2	[65,102]
OBP2	AT1G07640	AtDof1.1	[102,105]
TMO6	AT5G60200	AtDof5.3	[102,106]
HCA2	AT5G62940	AtDof5.6	[102,107]

In the table are reported the two PEAR proteins identified as mobile factors in root Protophloem Sieve Elements (PSE) and the four homologues identified as PSE-specifically or PSE-abundantly expressed *DOF* genes. OBP2 (ocs Binding Factor/OBF BINDING PROTEIN2), TMO6 (TARGET OF MONOPTEROS 6, HCA2 (HIGH CAMBIAL ACTIVITY2).

**Table 3 plants-09-00218-t003:** Arabidopsis DOF genes marked by H3K27me3.

Name	Gene ID	Synonymous [82]	References
DOF6	AT3G45610	AtDOF3.2	[65,102]
DAG1	AT3G61850	AtDOF3.7	[71,73,74]
DAG2	AT2G46590	AtDOF2.5	[72,79]
ITD1	AT4G00940	AtDOF4.1	[101]
PEAR1	AT2G37590	AtDOF2.4	[102]
PEAR2	AT5G02460	AtDOF5.1	[102,104]
TMO6	AT5G60200	AtDOF5.3	[102,106]
OBP2	AT1G07640	AtDOF1.1	[102,105]
HCA2	AT5G62940	AtDOF5.6	[102,107]
CDF6	AT1G26790	AtDOF1.3	[93,110,111]
OBP3	AT3G55370	AtDOF3.6	[98,112]
SCAP1	AT5G65590	AtDOF5.7	[113]
	AT1G28310	AtDOF1.4	
	AT2G28510	AtDOF2.1	[114]
	AT2G28810	AtDOF2.2	
	AT3G52440	AtDOF3.5	
	AT4G21030	AtDOF4.2	[91,105]
	AT4G21040	AtDOF4.3	[91]
	AT4G21050	AtDOF4.4	
	AT4G21080	AtDOF4.5	[91]
	AT4G24060	AtDOF4.6	[115]
	AT4G38000	AtDOF4.7	[116]

*DOF* genes that are enriched in the H3K27me3 epigenetic mark. Data from Bouyer et al. [108]. DAG1, DAG2 (DOF AFFECTING GERMINATION1, 2), ITD1 (INTERCELLULAR TRAFFICKING DOF1), SCAP1 (STOMATAL CARPENTER 1).

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
