# Peer review of "The DOF Transcription Factors in Seed and Seedling Development"

_plants, 2020, doi:10.3390/plants9020218_

Round 1

Reviewer 1 Report

In file

Author Response

In section 1:

- Reference 2 is wrong and should be replaced by the correct one.

            We are sorry for the mistake; it has been replaced

In section 2:

- I recommend to authors provide a more specific title to the section, as “DOF TFs regulate seed storage protein accumulation and mobilization”.

            The title of section 2 has been changed.

- In opinion of the reviewer, the author should first highlight the role of PBF, since is a DOF TF, purpose of this review. I strongly recommend that in text should first appear the description of this TF and then the author could provide a brief description of the interactor bZIP TF.

            The order of paragraphs has been inverted, in order to first describe the DOF TFs and then the bZip TFs.

- The reviewer recommends checking the reference 45. I am not totally sure is the correct one.

            Reference 45 was definitely wrong; it has been changed.

- The author should change the title of table including that this Tf could regulate hydrolase as well.

            According to the Reviewer suggestion, the title of the table has been changed in "DOF genes involved in seed storage protein accumulation and mobilization".

In section 3:

- I recommend to authors provide a more specific title to the section, as “Interaction DOF-DELLA represses seed germination”.

            According to the Reviewer suggestion, the title of section 3 has been changed.

- Line 117: it should be specified that is transcriptomic study, and somehow it should be explained that this study was performed in ga1-3 rga-t2 mutant seeds.

            According to the Reviewer suggestion, a further explanation of the study has been added: "A genome-wide comparative study allowed to identify genes specifically regulated by RGL2; this analysis compared the transcriptome of the ga1-3 rga-t2 seeds - which are not able to germinate - with that of ga1-3rga-t2rgl2-1, in which lack of RGL2 results in the rescue of seed germination".

- Line 146: it should be deleted the sentence “the first DOF gene for which an effect in plants was convincingly demonstrated”. Previous DOF studies are solid as well.

            The sentence has been modified "which was convincingly demonstrated to be involved in seed germination".

- Line 149: Sentence should be rewritten “DAG1 inactivation causes seed germination to require about a ten times lower GA concentration than wild-type to reach 50% germination [65]”. It is not too clear.

            The sentence has been modified as follows "dag1 mutant seeds require about a ten times lower GA concentration than wild type to reach 50% germination"

- Line 153: Sentence should be rewritten DAG1 repression of GA3ox1 is mediated by direct interaction with GAI, which is necessary for the binding of DAG1 to the DOF sites in the GA3ox1 promoter, as revealed by ChIP assay. It is not too clear.

            The sentence has been modified as follows "DAG1 cooperates with GAI to negatively regulate GA3ox1; indeed, GAI is necessary for the binding of DAG1 to the DOF sites in the GA3ox1 promoter, and it directly interacts with DAG1".

In section 4:

- Line 205: The author should clarify that only Red and Far Red light have a negative role in seed tolerance to deterioration.

            It has been specified.

Finally I really think that next articles should appear in this review, since are essential to complete an excellent review

            As suggested by the Reviewer the four articles have been added to the review.

Reviewer 2 Report

This is a satisfactory and compact review on DOF transcription factors (DOFs) mainly focused on their regulatory interactions in connection with seed development, dormancy and germination, as well as with (light-mediated) seedling development. This focus is appropriate for the contents of the Plants special issue on “Genetics of Seed Germination and Growth”. This article complements other, wider, recent reviews on DOFs (for example Noguero et al., 2013 http://dx.doi.org/10.1016/j.plantsci.2013.03.016; Gupta et al., 2015 https://link.springer.com/article/10.1007/s00425-014-2239-3; and Yanagisawa, 2016 https://doi.org/10.1016/B978-0-12-800854-6.00012-9), by citation and appropriate discussion of severalarticles published since these reviews, including research from the authors’ group that is relevant for this focused review. Compared with precedent reviews, I also found novel and interesting the pointed connection between DOFs and chromatin remodeling involving the repressive mark H3K27me3; with DOFs as potential gene targets of the Polycomb repressive complex 2 (see Table 3 and cited references 67 and 96-97). I only suggest some minor content and text corrections that may improve the manuscript:

1.- I recommend additional citation of the three recent reviews on DOFs mentioned above. Potential readers might find these wider-scope reviews useful and complementary.

2.- Besides red-light (perceived by Phytochrome fotoreceptors), blue-light (mainly perceived by theCryptochrome 1 fotoreceptor) is also important for seedling development in response to light. Indeed, DOFs modulate both Phytochrome-mediated and Cryptochrome 1-mediated light responses, the latter as shown for example by Ward et al., 2005 (this reference is cited in the manuscript -as reference number 101-, but only in connection to the information given in Table 3). I therefore, advice considering this for updating the information given in l. 175-190 on the involvement of DOFs in light responses.

3.- DOFs regulate the deposition in seeds of both storage proteins and lipid reserves. Perhaps the latter should be also mentioned in this article, despite the lesser knowledge on how DOFs impact on seed lipids, compared to the wider and deeper analyses on the effects of DOFs on seed storage proteins.

4.- Minor text corrections:

I would write the information given in lines 26-40 of the Introduction without the full stops at lines 36 and 40.

I would recommend that the legends of Figures 1 and 2 make reference to the original articles on which the depicted illustrations are based.

The listed references show inconsistent formatting, with only some of them including volume and page numbering information. The required, consistent, information should be used for all listed references.

Author Response

Reviewer 2

1.- I recommend additional citation of the three recent reviews on DOFs mentioned above. Potential readers might find these wider-scope reviews useful and complementary.

            The three reviews have been added.

2.- Besides red-light (perceived by Phytochrome fotoreceptors), blue-light (mainly perceived by the Cryptochrome 1 fotoreceptor) is also important for seedling development in response to light. Indeed, DOFs modulate both Phytochrome-mediated and Cryptochrome 1-mediated light responses, the latter as shown for example by Ward et al., 2005 (this reference is cited in the manuscript -as reference number 101-, but only in connection to the information given in Table 3). I therefore, advice considering this for updating the information given in l. 175-190 on the involvement of DOFs in light responses.

            According to the Reviewer suggestion, we have added a sentence on the effect of blue light in seedling development: "Besides phyA and phyB, the blue/UV-A absorbing cryptochromes (cry) have partially redundant functions in the control of photomorphogeneic responses, such as hypocotyl elongation and cotyledon expansion (for a review see Neff et al., 2000)."

In addition, we have added the following paragraph on the involvement of OBP3 in light-mediated seedling development:

" The Arabidopsis DOF protein OBF4 Binding Protein 3 (OBP3) has also been shown to be involved in light-dependent inhibition of hypocotyl elongation. Indeed, overexpression of OBP3 in the activation-tagged line sob1-D (suppressor of phyB-4 dominant) reverted the long hypocotyl phenotype of the phyB-4 mutant (Ward et al., 2005), suggesting that OBP3 promotes phyB-mediated inhibition of hypocotyl elongation. Interestingly, reduced OBP3 expression in the OBP3-RNAi lines displayed larger cotyledons, mainly under blue light, indicating that OBP3 acts downstream of both the phyB and cry1 photoreceptors, to repress cell expansion (Ward et al., 2005)."

3.- DOFs regulate the deposition in seeds of both storage proteins and lipid reserves. Perhaps the latter should be also mentioned in this article, despite the lesser knowledge on how DOFs impact on seed lipids, compared to the wider and deeper analyses on the effects of DOFs on seed storage proteins.

            According to the Reviewer suggestion, we have added a paragraph on DOF activity on lipid reserves:

" Lipids are also important seed storage compounds, mainly in oil crops such as soybean, maize and cotton. The soybean DOF proteins GmDOF4 and GmDOF11 have been shown to directly induce the acetyl CoA carboxylase and long-chain-acyl CoA synthetase biosynthetic genes [42]; consistently lipid and total fatty acids content was increased in GmDOF4 and GmDOF11 transgenic Arabidopsis seeds [42]. Similarly, overexpression of GhDOF1 from Gossypium hirsutum resulted in an increase of lipid levels in cotton seeds [43]."

In addition, following the suggestion of Reviewer 1, we have also added a paragraph on the activity of ZmDOF36 and ZmDOF6 on starch stored compound:

" Besides storage proteins, endosperm also contains starch, which is crucial for both seed yield and quality. The maize ZmDOF36 protein has been recently demonstrated to positively control starch accumulation [40]; indeed, ZmDOF36 overexpressing lines showed up-regulation of starch biosynthetic genes and increased starch content. In addition, six of these biosynthetic genes, namely ZmAGPS1a, ZmAGPL1, ZmGBSSI, ZmSSIIa, ZmISA1, and ZmISA3, were directly bound by ZmDOF36, in yeast one-hybrid assay [40]. Similarly, it was previously shown that knock-down of the endosperm-specific ZmDOF3 gene results in reduced starch content in endosperm of the ZmDOF3 RNAi transgenic lines [41]."

4.- Minor text corrections:

I would write the information given in lines 26-40 of the Introduction without the full stops at lines 36 and 40.

            Done.

I would recommend that the legends of Figures 1 and 2 make reference to the original articles on which the depicted illustrations are based.

            References have been added.

The listed references show inconsistent formatting, with only some of them including volume and page numbering information. The required, consistent, information should be used for all listed references.

            We are sorry for incorrectly formatted references; in this revised version, we have carefully checked the listed references.

Round 2

Reviewer 1 Report

Ready to be published.